# DNA Vaccine Administered by Cationic Lipoplexes or by In Vivo Electroporation Induces Comparable Antibody Responses against SARS-CoV-2 in Mice

**DOI:** 10.3390/vaccines9080874

**Published:** 2021-08-06

**Authors:** Allegra Peletta, Eakachai Prompetchara, Kittipan Tharakhet, Papatsara Kaewpang, Supranee Buranapraditkun, Teerasit Techawiwattanaboon, Tayeb Jbilou, Pratomporn Krangvichian, Sunee Sirivichayakul, Suwimon Manopwisedjaroen, Arunee Thitithanyanont, Kanitha Patarakul, Kiat Ruxrungtham, Chutitorn Ketloy, Gerrit Borchard

**Affiliations:** 1Section of Pharmaceutical Sciences, Institute of Pharmaceutical Sciences of Western Switzerland (ISPSO), University of Geneva, 1211 Geneva, Switzerland; allegra.peletta@unige.ch (A.P.); tayeb.jbilou@unige.ch (T.J.); 2Department of Laboratory Medicine, Faculty of Medicine, Chulalongkorn University, Bangkok 10330, Thailand; Eakachai.p@chula.ac.th (E.P.); kankittipan12@hotmail.com (K.T.); bsuprane2001@yahoo.com (S.B.); Sunee.S@chula.ac.th (S.S.); 3Center of Excellence in Vaccine Research and Development (Chula Vaccine Research Center, Chula VRC), Faculty of Medicine, Chulalongkorn University, Bangkok 10330, Thailand; Kaewpangpapatsara@gmail.com (P.K.); teerasit.kku@gmail.com (T.T.); Pratomporn88@gmail.com (P.K.); kpatarakul@gmail.com (K.P.); rkiatchula@gmail.com (K.R.); 4Thai Pediatric Gastroenterology, Hepatology and Immunology (TPGHAI) Research Unit, Faculty of Medicine, Chulalongkorn University, Bangkok 10330, Thailand; 5Department of Microbiology, Faculty of Medicine, Chulalongkorn University, Bangkok 10330, Thailand; 6Multidisciplinary Program of Medical Microbiology, Graduate School, Chulalongkorn University, Bangkok 10330, Thailand; 7Department of Microbiology, Faculty of Science, Mahidol University, Bangkok 10400, Thailand; swiboonut@gmail.com (S.M.); arunee.thi@mahidol.edu (A.T.)

**Keywords:** DNA vaccines, SARS-CoV-2, cationic liposomes, lipoplexes, immunogenicity

## Abstract

In view of addressing the global necessity of an effective vaccine in the SARS-CoV-2 pandemic, a plasmid DNA vaccine, expressing for the spike (S) protein and formulated in lipoplexes, was manufactured and tested for in vitro transfection and in vivo immunogenicity. Blank cationic liposomes of 130.9 ± 5.8 nm in size and with a zeta potential of +48 ± 12 mV were formulated using the thin-film layer rehydration method. Liposomes were complexed with pCMVkan-S at different N/P ratios. Ratios of 0.25:1 and 1:1 were selected according to their complex stability and controlled size compared to other ratios and tested in vitro for transfection studies and in vivo for immunogenicity. Both selected formulations showed enhanced neutralizing antibody responses compared to pCMVkan-S injected alone, as well as an increased T cell response. The titers observed were similar to those of intramuscular electroporation (IM-EP), which was set as an efficacy goal.

## 1. Introduction

Severe acute respiratory syndrome coronavirus 2 (SARS-CoV-2) is a positive-sense single-stranded RNA virus from the coronavirus family that causes a respiratory disease called COVID-19 in humans [1]. The rapid spread of the disease and the gravity of the symptoms caused in some patients, resulting in strain on healthcare systems worldwide, led the World Health Organization (WHO) to declare COVID-19 a pandemic on 11 March 2020 [2]. The COVID-19 pathogen is suggested to be a bat-borne virus that was probably amplified in an intermediate host before transitioning to humans. The spillover event causing this reservoir transition likely occurred in late 2019 [3]. The sudden spread of the disease has demanded rapid development of therapeutics and prophylactic intervention in a short period of time. Since early 2020, huge efforts have been made in vaccine research. According to WHO, there are currently more than 100 COVID-19 vaccine candidates under clinical evaluation [4]. Among these, around 10 vaccines were approved by emergency use authorization (EUA) or by standard procedure in several countries [5].

Several SARS-CoV-2 target antigens as well as several types of vaccines are under investigation. One of the most suitable antigens to elicit protective immunity appears to be the SARS-CoV-2 spike (S) protein, a glycoprotein expressed on the viral surface. It mediates the viral entry into respiratory epithelial cells through interaction with the angiotensin-converting enzyme 2 (ACE-2) receptor [6,7]. In fact, the SARS-CoV-1 S protein had already been shown in 2003 to trigger the production of serum neutralizing antibodies, inducing protective immunity upon virus challenge [8]. The S protein was shown to be a valid antigen also for SARS-CoV-2 vaccine design, triggering specific antibody production and T cell response in early clinical trials [9,10].

Among the various types of vaccines, gene-based vaccination using plasmid DNA and mRNA raised major interest. DNA vaccination is a technique where a plasmid is genetically engineered by adding the sequence expressing for the antigen of interest. In such manner, once delivered to target cells, the plasmid will activate antigen production using the host cell machinery, thereby inducing antigen presentation and recognition that will trigger the adaptive immune system [11]. This technique holds several advantages, including cost-effectiveness, stability during storage, activation of both cellular and humoral immune responses, and no risk of infection [12,13]. In contrast to mRNA vaccines, DNA vaccines do not require an in vitro transcription step, and are more stable than mRNA vaccines, which currently still require ultra-low temperatures for long-term storage [14]. These advantages may significantly decrease costs of production and logistics of distribution. This would increase vaccine accessibility for countries that have been insufficiently supplied with SARS-CoV-2 vaccines until now. However, two major limitations are presented by plasmid vaccination, including low immunogenicity and poor transfection efficiency when the naked plasmid DNA is used [13]. DNA vaccines were already shown to be able to induce immunity and protection in animal models against both SARS-CoV-1 and SARS-CoV-2 infections [15,16]. Recently, a SARS-CoV-2 DNA vaccine has been investigated in a phase I clinical trial [17].

However, these studies employed in vivo electroporation as the method of administration, which required specific equipment such as an electric field generator and specific probes that may not be practical for large scale vaccination campaigns. Plasmid DNA delivery may alternatively be improved by using suitable delivery systems and/or adjuvants. The delivery system needs to deliver the plasmid DNA preferably to antigen presenting cells (APCs) at the site of injection, enhancing the elicited immune response and reducing toxicity. Among the various delivery systems investigated, cationic liposomes have been extensively used for gene-based vaccine delivery thanks to their stability, enhanced transfection efficiency, protection of payload from degradation, and intrinsic immunogenicity [18,19,20].

We previously reported a SARS-CoV-2 DNA vaccine candidate expressing spike (S) protein delivered by intramuscular electroporation (IM-EP) and showing excellent immune response in mice [21]. As in vivo electroporation is an expensive technique and requires professional personnel, it might not be adapted to a pandemic situation. In this study, we aimed to investigate the immunogenicity in mice of a SARS-CoV-2 DNA vaccine candidate expressing S protein formulated for intramuscular (i.m.) injection using DOTAP-based cationic liposomes (DOTAP4).

## 2. Materials and Methods

For all experiments, the lipids 1,2-dioleoyloxy-3-trimethylammoniumpropane (DOTAP) chloride, (Mw: 698.54 g/mol) and 1.2 dioleoyl-sn-glycero-3-phosphoethanolamine (DOPE) were obtained from Sigma Aldrich (Buchs, Switzerland), whereas dipalmitoylphosphatidylcholine (DPPC) was purchased from Avanti Polar Lipids (Alabaster, AL, USA). Methanol and isopropanol were purchased from Fisher Scientific (Loughborough, UK), Chloroform Emprove® from Merck (Darmstadt, Germany). Phosphate Buffered Saline (PBS) Mg-/Ca- (Gibco®, Thermofisher, Carlsbad, CA, USA) was used for all experiments. AI334 Anti-Spike protein antibody was purchased from the antibody facility of the University of Geneva (Geneva, Switzerland). FITC-labeled secondary rabbit anti-mouse antibody was obtained from Dako (Glostrup, Denmark). 

### 2.1. Cell Lines

Human embryonic kidney 293 cells (HEK 293, ATTC CRL-1573, LGC Standards, Teddington, UK) for transfection studies were cultured in Modified Eagle’s Medium (MEM) purchased from Gibco, Thermofisher (Carlsbad, CA, USA) which, in order to obtain complete MEM (C-MEM), was supplemented with 10% fetal calf serum (FCS), 0.1% penicillin-streptomycin (Pen/Strep), 1 mM sodium pyruvate, and 0.1% non-essential amino acids (all obtained from Gibco, Thermofisher, Carlsbad, CA, USA) at 37 °C and in an atmosphere containing 5% CO_2_. Murine macrophage RAW 264.7 cell line (ATTC TIB-71, Manassas, VA, USA) used for toxicity assessment was cultured in Dulbecco’s Modified Eagle’s Medium (DMEM) purchased from Gibco, Thermofisher (Carlsbad, CA, USA), which, in order to obtain complete DMEM (C-DMEM), was supplemented with 10% FCS, 0.1% Pen/Strep, sodium pyruvate 1 mM, and glutamine 1 mM (all obtained from Gibco, Thermofisher, Carlsbad, CA, USA) under the same conditions as mentioned above. African green monkey kidney epithelial cells (Vero-E6, ATCC CRL-1586™, Manassas, VA, USA) were cultured in DMEM 2% fetal bovine serum (FBS) supplement with 0.1% Pen/Strep.

### 2.2. Plasmid DNA Construct

The details of plasmid DNA construction were described in the previous report [21]. In brief, the humanized codon of cytoplasmic-deleted SARS-CoV-2 spike protein was synthesized by GeneScript (Piscataway, NJ, USA), then subcloned into pCMVkan expression vector, designated as pCMVkan-S. pCMVkan-S was propagated in *E. coli* DH5-alpha (Invitrogen, Carlsbad, CA, USA) and purified by Qiagen endotoxin-free giga plasmid kit (Hilden, Germany) following the manufacturer’s protocol. Characterization of the plasmids was performed by nucleotide sequencing and gel electrophoresis.

### 2.3. Mice Experiments

Female ICR mice at 6 weeks of age (weight 20–25 g), procured from the National Laboratory Animal Center, Mahidol University, were randomly allocated into 4 groups with 5 mice/group. One hundred micrograms of pCMVkan-S were injected intramuscularly (IM) 3 times at 2-week interval (weeks 0, 2, and 4) using 4 different conditions including group 1: naked pCMVkan-S injection (IM-naked), group 2: pCMVkan-S formulated with DOTAP4 liposomes at an N/P ratio of 1:1, group 3: pCMVkan-S formulated with DOTAP4 liposomes at an N/P ratio of 0.25:1, and group 4: pCMVkan-S immunized via IM with electroporation (IM-EP) using TriGrid delivery system (Ichor Medical System, San Diego, CA, USA) [22]. Blood samples were collected every 2 weeks after immunization designated as weeks 0, 2, 4, and 6. At week 6, mice were euthanized by 30% CO_2_ inhalation, and splenocytes were collected for T cell response analysis. In addition, bronchoalveolar lavage (BAL) was performed on the euthanized mice by inflating the lung with 1 mL of 2% BSA in PBS twice.

### 2.4. Liposomal Formulation and pCMVkan-S Complexation

Liposomes were formulated using the thin film rehydration method described elsewhere [23]. Lipids DPPC, DOPE, and DOTAP were dissolved in 4 mL chloroform:methanol [9:1] at a ratio of DPPC:DOPE:DOTAP [8:4:4], and the solvent was evaporated under vacuum at 40 °C (water bath) using a Rotavapor (V-855; Büchi, Flawil, Switzerland) for 1 h. A thin film of lipids formed and was successively rehydrated with 2 mL warm phosphate buffered saline (PBS). After rehydration, the sample was sonicated for 12 min, resulting in small unilamellar vesicle (SUV) formation. Liposomal zeta potential, size, and size distribution were measured using a Nano ZS Zeta-Sizer (Malvern, Malvern, UK) by dynamic light scattering (DLS) at a scattering angle of 173° using a He-Ne laser beam at λ = 633 nm, a refractive index of 1.345, and absorbance of 0.010 at 25 °C. Measurements were conducted in disposable low-volume polystyrene cuvettes, and samples were left to equilibrate for 20 s before measurements. Each measurement constituted a minimum of 10 runs of 10 s each. For plasmid-liposome complexation, a suitable volume of liposome suspension calculated on the basis of the various positively charged nitrogen to negatively charged phosphate molar ratios (N/P ratios), from 0.25:1 to 100:1, was diluted in PBS, and plasmid was subsequently added dropwise. The solution was mixed by trituration and let rest for 30 min at room temperature, upon which size and zeta potential values were determined.

### 2.5. U-HPLC Quantification of DOTAP

DOTAP concentration was measured in order to quantify the concentration of liposomes by a U-HPLC protocol adapted from previous studies [24]. Briefly, a Waters Acquity system (Milford, MA, USA) equipped with a binary solvent delivery pump, sample manager, auto sampler, column-heating compartment, and a photo diode array (PDA) detector was used. Separation was carried out on an XBridge (London, UK) C18, 3.5 µm, 2.1 × 100 mm column maintained at 40 °C. The system was equipped with Empower 2 software for data acquisition and handling. Lipids were eluted using binary linear gradients starting from a mixture of 50% A and 50% B to 100% B in 1.88 min, followed by a 2.08 min plateau at 100% B, where A was 0.15% (*v*/*v*) trifluoracetic acid (TFA) in water and B was 0.05% (*v*/*v*) TFA in isopropanol. The mobile phase composition was then changed back to the initial solvent mixture. The flow rate of the mobile phase was set at 0.2 mL/min. The samples for injection were diluted directly in isopropanol. The calibration curve was established with DOTAP standards from 1 mg/mL to 0.015 mg/mL at 1:1 dilution steps. UV detection was performed at a wavelength of 205 nm using a PDA detector. Concentration of DOTAP was calculated by measuring the area under the curve (AUC).

### 2.6. Transmission Electron Microscopy (TEM)

Carbon coated grids (Quantifoil, Hatfield, PA, USA) were glow-discharged using an EMS GlowQube instrument. The samples were then applied to the grids at 0.02–0.04 mg/mL, incubated for 1 min, and blotted. The grids were then washed and stained sequentially with PBS and 2% uranyl acetate and finally air-dried. The micrographs were acquired using a G2 Sphera microscope (Thermofisher, Waltham, MA, USA) operating at 120 kV with a defocus range of −1 to −2 µm. Images were analyzed with Fiji software (ImageJ, 1.53c plus).

### 2.7. Mitochondrial Activity (WST-1)

RAW 264.7 cells were plated at an initial density of 1 × 10^5^ cells/mL in C-DMEM in a 96-well plate and incubated at 37 °C, 5% CO_2_. The next day, 100 µL of cationic liposomes at decreasing concentrations, from 7 mg/mL to 0.22 mg/mL, were added to the cells and incubated for 4 h. Medium was aspirated, and WST-1 reagent (4-[3-(4-iodophenyl)-2-(4-nitrophenyl)-2H-5-tetrazolio]-1,3-benzene disulfonate; Roche, Basel, Switzerland) diluted at a ratio of 1:10 in C-DMEM was added to the cells and incubated 30 min at 37 °C and 5% CO_2_. UV absorbance of the samples was then measured at 450 nm on a plate reader (Biotek, Synergy Mx, Winooski, VT, USA).

### 2.8. Liposomal Complex Stability

Liposome-plasmid complexes were mixed with DNA loading dye^®^, and 20 µL of each sample was placed in a 1% agarose gel stained with 8 µL of SYBR^®^ safe DNA stain (Invitrogen, Carlsbad, CA, USA). A 1 kbp DNA ladder (Thermofisher, GeneRuler^®^, Waltham, MA, USA,) was run in parallel on each gel, and plasmid (pCMVkan-S) alone served as a control. A voltage of 80 V was applied for 30 min using Tris-Borate-EDTA (TBE) as a buffer. Plasmid migration was visualized by UV transillumination (Bio-Rad, ChemiDoc^®^ XRS, Hercules, CA, USA).

### 2.9. Transfection Studies and Immunofluorescence

HEK293 adherent cells (plated in C-MEM without Pen/Strep), cultured on glass coverslips and transfected 24 h before the experiment with pCMVkan-S/liposome complexes, were used to detect the specific viral protein. Complexes of Lipofectamine 2000 (Invitrogen, Carlsbad, CA, USA) and pCMVkan-S were prepared following the manufacturer’s protocol and used as positive control. pCMVkan-S alone and non-transfected cells were used as negative control groups. Transfected HEK293 cells were fixed with PBS + 4% formaldehyde (Reactolab, Servion, Switzerland) for 30 min at ambient temperature and blocked with PBS + 40 mM ammonium chloride (NH_4_Cl) for 5 min. Cells were permeabilized in PBS + 0.1% Triton X-100 for 15 min, incubated in PBS + 0.2% BSA (PBS-BSA) (Applichem, Darmstadt, Germany) for 5 min, and then incubated with an anti-spike protein antibody (Anti-S AI334, University of Geneva, Geneva, Switzerland) at a concentration of 5 ng/mL in PBS-BSA for 1 h at 37 °C. After 3 washes with PBS-BSA, cells were incubated with a FITC-labeled secondary rabbit anti-mouse antibody diluted 1:100 in PBS-BSA for 1 h at 37 °C. After 3 washes, cells were stained with 1:2000 Hoechst dye (Invitrogen, Carlsbad, CA, USA) in PBS for 10 min at ambient temperature protected from light, then washed again 3 times [25]. Cells were mounted on slides with Vectashield (Vectorlab, Burlingame, CA, USA). Images were obtained using a Nikon (Egg, Switzerland) A1r Spectral point scanning confocal microscope with a 40× oil immersion objective.

### 2.10. Enzyme-Linked Immunosorbent Assay

End-point titers of IgM, IgA, total IgG, IgG1, and IgG2a subclasses in sera as well as total IgG and IgA in BAL from immunized mice were measured by ELISA using a method described previously [21]. In brief, MaxiSorp™ flat-bottom 96-well plates (Nunc, Roskilde, Denmark) were coated with 100 ng of S protein trimer (Acro Biosystems, Newark, DE, USA) diluted in coating buffer (0.1 M sodium carbonate) pH 9.5 and incubated overnight at 4 °C. Plates were washed 5 times with PBS containing 0.05% Tween-20 (PBS-T), followed by blocking with 1% bovine serum albumin (BSA) in PBS-T for 1 h at 37 °C. After washing, plates were incubated with a 5-fold serial dilution of mouse sera (25 µL) starting from 1:20 to 1:125,000 and incubated for 1 h at 37 °C. The plates were then washed and incubated with 1:5000 dilution of horseradish peroxidase (HRP)-conjugated secondary anti-bodies including rabbit anti-mouse IgG (KPL, USA), -IgM, -IgA, -IgG1, or -IgG2a (BioLegend, San Diego, CA, USA) for an additional 1 h at 37 °C. After washing, 100 µL of tetramethylbenzidine (TMB) substrate (BioLegend, San Diego, CA, USA) was added and incubated for 5 min. The reactions were then stopped with 50 µL of 0.16 N sulfuric acid. The absorbance was measured at a wavelength of 450 nm using a Varioskan microplate reader (ThermoFisher Scientific, Vantaa, Finland). End-point titers were determined and expressed as the reciprocals of the final dilution that emitted an optical density exceeding 4 times of the background (BSA plus secondary antibody).

### 2.11. Micro-Neutralization Test (MN)

MN for the measurement of neutralizing antibody titers was performed at a certified biosafety level 3 facility, Department of Microbiology, Faculty of Science, Mahidol University, Thailand. The procedure of the MN assay was described previously [26]. In brief, heat-inactivated immunized mice sera were 2-fold serially diluted in DMEM supplemented with 2% FBS (DMEM + 2% FBS), starting from 1:10. The diluted sera were then mixed with equal volumes of 100-times of TCID50 [50% tissue culture infectious dose) of the SARS-CoV-2 virus isolated from a confirmed COVID-19 patient at Bamrasnaradura Infectious Diseases Institute, Nonthaburi, Thailand (SARS-CoV-2/01/human/Jan2020/Thailand) and incubated for 1 h at 37 °C. The serum-virus mixtures were transferred into pre-seeded Vero-E6 cells (1 × 10^4^ cells/well, duplicates). After 2 days of infection, cells were fixed and permeabilized with ice-cold 1:1 methanol:acetone for 20 min at 4 °C. The fixed cells were washed 3 times with PBS-T, then soaked with blocking buffer (1× PBS containing 2% BSA and 0.1% Tween 20) for 1 h. The plates were then washed 3 times with PBS-T. For detection of viral antigen, the SARS-CoV/SARS-CoV-2 nucleocapsid mAb (40143-R001; Sino Biological, Wayne, PA, USA) at a dilution of 1:5000 in 1× PBS containing 0.5% BSA and 0.1% Tween 20 was added to each well and incubated for 1 h at 37 °C. The plates were then washed 3 times with PBS-T and a secondary antibody, HRP-conjugated goat anti-rabbit polyclonal antibody (P0448; Dako, Glostrup, Denmark) at a dilution of 1:2000, was added. After incubation for 1 h at 37 °C, the plates were washed 3 times and the substrate, SureBlue TMB 1 component microwell peroxidase substrate (SeraCare Life Sciences, Milford, MA, USA), was added in each well (50 µL/well) and incubated at room temperature in the dark for 10 min. The reaction was then stopped with 50 µL of 1 N HCl. The absorbance was measured at 450 nm (A450) and 620 nm (A620) as a reference wavelength with an ELISA plate reader. The average A450/620 was determined for quadruplicate wells of 100TCID50 and negative control wells (CC), and a neutralizing endpoint was determined by using a 50% specific signal calculation. The endpoint titer was expressed as the reciprocal of the highest dilution of serum with average A450/620 value (duplicate wells) less than X, where X = [(average A450/620 of 100TCID50 wells) − (average A450/620 of CC wells)]/2 + (average A450/620 of CC wells). Serum samples that tested negative at a dilution of 1:10 were assigned an MN50 titer of 10.

### 2.12. Mouse T Cell Response Analysis by IFN-γ ELISPOT and Cytokine Release Assays

The procedure of mouse IFN-γ ELISPOT and cytokine release assays used in this study was described in our previous report [21]. Briefly, 96-well nitrocellulose membrane plates (MAHA S45; Merk Millipore, Darmstadt, Germany) were coated with 50 µL/well of 10 µg/mL of anti-mouse IFN-γ (AN18) monoclonal antibody, mAb (Mabtech, Nacka Strand, Sweden) at 37 °C with 5% CO_2_ for 3 h. Then, the plates were washed 6 times with PBS and blocked with 200 µL R10 medium (RPMI + 10% FBS)/well for at least 1 h at room temperature. Mouse splenocytes at 5 × 10^5^ cells/well were cultured with SARS-CoV-2 spike peptide pools spanning the entire sequence of spike protein, 25 peptides/pool (Mimotopes, Mulgrave, Victoria, Australia) at a final concentration of 2 µg/mL at 37 °C, 5% CO_2_ for 40 h. Pools 1–5 and 6–10 corresponded to S1 and S2 regions of spike protein, respectively. Culture medium alone and concanavalin A (ConA) served as negative and positive controls, respectively. After incubation, the plates were washed and incubated with 1 µg/mL of anti-mouse IFN-γ-biotinylated mAb (R4-6A2 biotin; Mabtech, Nacka Strand, Sweden) in PBS for 3 h at room temperature. After washing, the plates were incubated with 50 µL/well of streptavidin-alkaline phosphatase (Mabtech, Nacka Strand, Sweden) diluted 1:1000 in PBS for 1 h. The plates were washed, then 100 µL of the substrate solution [5-bromo-4-chloro-3-indolyl-phosphate/nitro blue tetrazolium; BCIP/NBT) were added into each well. The reaction was stopped by washing extensively in tap water. The plate was left to dry, then the spots were counted using an ELISpot reader (ImmunoSpot^®^ Analyzer, Cleveland, OH, USA). Results were expressed as spot-forming cells (SFCs)/106 splenocytes after subtraction of the spots from negative control wells. In parallel, cytokines released from mouse splenocytes stimulated with spike pooled peptides (pools 1–10) were also analyzed. After 40 h of peptide stimulation, the culture supernatant was collected and analyzed for cytokine secretion using mouse Th1/Th2 Cytokine Panel (Biolegend, San Diego, CA, USA). The signals of each cytokine were analyzed by flow cytometry (BD FACSCalibur, Becton Dickinson, San Jose, CA, USA).

### 2.13. Statistical Analysis

Statistical analysis was performed using GraphPad Prism 8 software. Comparisons of the data between groups were made using two-way ANOVA test. In vivo results comparisons of the data between groups were made using Mann–Whitney tests. All *p* values < 0.05 were defined as statistically significant.

## 3. Results

Obtained liposomal formulations were characterized for physicochemical parameters decisive for their efficacy and safety, as well as for their ability to complex pDNA. Stability of these complexes in biological buffer was determined. Transfection of HEK 293 and expression of spike protein was assessed in vitro. The ability of the liposomal formulations, in comparison to the “naked” DNA plasmid injected or applied by intramuscular electroporation, to elicit specific immune response was assessed in vivo.

### 3.1. Characterization of Blank Liposomes

DPPC:DOPE:DOTAP 8:4:4 (in the following called DOTAP4) blank liposomes had a size (intensity based z-average) of 130.9 ± 5.8 nm, with a PDI of 0.210 ± 0.028. Liposomes showed a zeta potential of +48 ± 12 mV, with the positive charge allowing complexation with negatively charged plasmids by electrostatic interaction. TEM performed on blank liposomes revealed particle sizes close to those observed by DLS. However, sample preparation and drying appeared to flatten liposomes, making them appear slightly larger (Figure 1A, red arrow).

WST-1 toxicity results showed absence of toxicity on RAW 264.7 cells after 4 h incubation and up to a concentration of 7 mg/mL of blank liposomes (Figure 2). An incubation time of 4 h was chosen as liposomes are estimated to be taken up by the cells during this time period. Data were normalized to results obtained with non-treated cells representing 100% of mitochondrial activity. For all sample conditions, mitochondrial activity was >79%. This difference was not significant when compared to the non-treated sample, showing the absence of cytotoxicity of the samples at high concentrations. As observed in Figure 2, incubation with SDS 1% (positive control) resulted in complete loss of mitochondrial activity.

### 3.2. DOTAP Quantification

In order to quantify the exact liposomal content in suspensions, a U-HPLC method was developed to quantify the concentration of DOTAP present in the sample (Figure 3). The standard curve of DOTAP was measured from 1 mg/mL to 0.015 mg/mL (R^2^ = 0.99). The limit of detection (LOD) was 0.0042 µg/mL of DOTAP, and the limit of quantification (LOQ) was 0.0126 µg/mL. Liposome concentration was calculated by first measuring the AUC of the corresponding DOTAP peak and then extrapolating the total concentration of the sample by using a known liposomal ratio. Concentrations of liposomes between 5 and 22 mg/mL were calculated, corresponding to yields varying from 25 to 100% depending on the thin film method preparation outcome.

### 3.3. Liposome/pCMVkan-S Complexation

Several N/P ratios from 0.25:1 to 100:1 were tested and characterized by DLS for size and polydispersity, zeta potential for charge, TEM for size and shape, and electrophoretic mobility for complex stability. N/P ratios from 0.25:1 to 5:1 showed a size in the nanometer range and some aggregation, with a PDI of around 0.4. Despite this, no significant difference was observed compared to blank liposomes. Increasing concentrations of liposomes, translating to a higher N/P ratio, corresponded to increased aggregation of the lipoplexes (Figure 4). Above a 10:1 N/P ratio, visible aggregates were observed in the sample, and polydispersity was consistent. On the other hand, an N/P ratio of 100:1 resulted again in smaller sizes, but in a PDI value of 0.723 ± 0.023 nm, which showed the sample to be heterogeneous. Our aim was to use particles having a size smaller than 1 µm, which was shown to enhance APC uptake [27].

With regard to zeta potential values, we generally observed an excess of pCMVkan-S with negatively charged particles. The highest ratio [100:1] corresponded to a zeta potential switch to positive values (Figure 5). N/P ratios of 25:1 and 50:1 showed a zeta potential of −19.42 ± 3.48 mV and +15.25 ± 0.63 mV, respectively, and corresponded to the two samples closer to a neutral charge. Often a neutral charge corresponds to an increased aggregation in the sample, which is what was observed in the size results. A large excess of liposomes or nucleic acid is expected to reduce aggregation.

Although no toxicity was observed in Figure 2, WST-1 assay on RAW 264.7 cells treated with blank liposomes, a negatively charged formulation is preferred over a positively charged sample with similar characteristics, as positive charges tend to destabilize cell membranes, thus causing toxicity. In Figure 6, the behavior of the complexes in PBS at various N/P ratios was determined by electrophoretic mobility. The aim was to determine the degree of complexation at each N/P ratio. Retention of pCMVkan-S in the gel in the presence of phospholipids was observed for all samples. DNA retention was increasingly enhanced at higher N/P ratios; complete retention, at a 25:1 N/P ratio, corresponded to the appearance of visible aggregates, which correlated to a certain sample instability. N/P ratios of 0.25:1, 1:1, and 5:1 showed the highest amounts of free DNA, but their size and zeta potential profiles were shown to be more favorable. Therefore, 0.25:1 and 1:1 ratios were selected for further studies as they presented smaller size, lower PDI, and no significant difference with blank liposome sample, and nevertheless showed DNA complexation in electrophoresis studies (Figure 5). Liposome/pCMVkan-S complexation characterization, TEM images of blank liposomes, and DNA-lipoplexes are shown in Figure 1. After pCMVkan-S addition, some aggregation was observed for N/P ratios of 1:1 and 0.25:1, as predicted by DLS. Shrunken shapes in Figure 1B,C confirmed liposome DNA complexation. The dark shade around the particles was DNA in excess, whose presence was confirmed in Figure 5.

Electrophoretic mobility in C-MEM was tested to assess lipoplex stability in a biologically relevant medium. Non-complexed “naked pCMVkan-S” was degraded completely after 15 min. As shown in Figure 7A, pCMVkan-S retention in the wells was observed as well as a large excess of pCMVkan-S (as expected from zeta potential results). Next, we tested the stability in C-MEM at 37 °C for 24 h (Figure 7B). Liposomes still retained and protected complexed pCMVkan-S, as DNA signal was observed at baseline for both ratios.

### 3.4. Intracellular S Protein Expression

HEK293 cells were successfully transfected with liposome/pCMVkan-S complexes (Figure 8). The amount of fluorescent protein appeared as pronounced as the positive control, lipofectamine/pCMVkan-S complexes. No fluorescence was observed for both non-treated cells and non-transfected cells treated with FITC secondary antibody only.

### 3.5. Immunogenicity in Mice

#### 3.5.1. Binding Antibodies

Immunized mouse sera were serially collected for total S-specific IgM, IgA, IgG, and IgG subclass measurement. The results revealed that S-specific IgM was detected after the first dose (Wk2) of vaccine. The endpoint IgM titers were comparable in all groups (Appendix A). IgG was detected at low levels after the first dose, and the titers were significantly increased when the second dose was applied. Interestingly, only slight increases in IgG titers were observed after the third dose. The titers induced by both ratios of DOTAP4/pCMVkan-S and IM-EP were significantly higher (>10-fold) than those induced by naked plasmid injection (Figure 9A). Mice immunized with both ratios of DOTAP4/pCMVkan-S induced IgG titers comparable to IM-EP at all time-points. At week 6, mean endpoint titers of total IgG for IM-naked, N/P 1:1, N/P 0.25:1, and IM-EP groups were 163, 1804, 1996, and 1681, respectively. Importantly, the analysis of IgG subclasses IgG2a and IgG1, which reflect Th1 and Th2 responses, respectively, revealed comparable titers of IgG2a and IgG1 in all groups (Figure 9B). Total IgG was also analyzed in BAL at week 6. Consistent with serum IgG, although statistically not significant, the IM-naked group showed the lowest IgG titer, whereas both groups of DOTAP4/pCMVkan-S induced BAL IgG levels comparable to the IM-EP group. Means of endpoint BAL IgG titers for IM-naked, N/P 1:1, N/P 0.25:1, and IM-EP groups were 129, 285, 312, and 280, respectively (Appendix A). Unfortunately, spike protein-specific IgA in serum and BAL were undetectable in all groups (data not shown).

#### 3.5.2. Neutralizing Antibodies

Immunized mouse sera were subjected to analysis of neutralization activity against wild-type SARS-CoV-2 virus. Consistent with IgG results, neutralizing antibody levels were detected in most of the immunized animals at 2 weeks after the first doses (Figure 10). The titers gradually increased following the second immunization (week 4). At this time-point, the MN50 geometric mean titers (GMTs) in the IM-naked group were at least 2.6-fold lower than in other groups; the GMTs for IM-naked, N/P 1:1, N/P 0.25:1, and IM-EP groups were 243, 970, 1280, and 640, respectively. At week 6, the titers in the IM-naked group were significantly lower than in other groups. Importantly, at this time-point, both groups of DOTAP4/pCMVkan-S showed comparable levels of GMTs, with the IM-EP group used as a positive control group. The GMTs at week 6 for IM-naked, N/P 1:1, N/P 0.25:1, and IM-EP groups were 557, 5120, 5120, and 5881, respectively.

#### 3.5.3. T Cell Response

We measured IFN-γ secreting cells from splenocytes of immunized mice using ELISpot assay at 2 weeks after the third immunization. In all groups, mouse splenocytes secreted IFN-γ when stimulated with spike peptide pools (Figure 11). The highest magnitude of the IFN-γ ELISpot response was observed in the IM-EP group (mean = 2572 SFC/106 splenocytes), which showed significantly higher SFC than other groups. Interestingly, both DOTAP4/pCMVkan-S groups showed significantly higher magnitudes of response than the naked pCMVkan-S group (IM-naked group). An N/P ratio of 0.25:1 tended to induce higher responses than those of N/P 1:1 (1.7-fold increase), however, at *p* > 0.05. The average SFCs for IM-naked, N/P 1:1, and N/P 0.25:1 groups were 330, 738, and 1239 SFC/106 splenocytes, respectively. Of note, the IM-naked group showed the lowest response when compared to other groups. Cytokine secretion in culture supernatant after stimulation of mouse splenocytes with spike peptide pools revealed that IL-2, a marker of Th1 response, was the highest induced by the IM-EP group followed by the N/P 0.25:1 group. More importantly, IL-4, a Th2 marker, was not detected in any group (Appendix A).

## 4. Discussion

Several vaccines have been approved for emergency use in several countries: for example mRNA vaccines mRNA-1273 (Moderna, MA, USA) and BNT162b2 (Pfizer/BioNTech, NY, USA); viral vectors ChAdOx1 nCoV-19 (Oxford/AstraZeneca, Oxford and Cambridge, UK) and Sputnik V (Gamaleya Research Institute, Moscow, Russia); and inactivated virus CoronaVac (Sinovac, Beijing, China) [5]. However, high demand for vaccine remains. To achieve herd immunity, it was estimated that approximately 70% of the world population (3.7 billion individuals) must be vaccinated [28]. Moreover, annual or seasonal COVID-19 vaccination might be required. Thus, efforts to develop sufficient and affordable COVID-19 vaccines for the world population were authorized.

In this study, we successfully developed lipid nanoparticles based on DOTAP that may be used as an alternative DNA vaccine delivery system. The DNA vaccine candidate used in this study was a plasmid DNA expressing full-length spike (pCMVkan-S) and selected according to the superiority of its immunogenicity profile compared to truncated S1 or S2 spike proteins [21]. In our previous study, pCMVkan-S was delivered by in vivo electroporation, which required a special device, probes, and well-trained personnel. Thus, the usage of electroporation might be limited in some circumstances, such as in a pandemic and resource-limited area. In this regard, the Swiss Federal Institute of Intellectual Property was contacted in order to identify a formulation with freedom to operate and containing low-cost and easy-to-handle excipients. DOTAP4 was selected for pCMVkan-S, as its development would be advantageous on several levels in a pandemic setting.

As apparent, one of the main issues for protection of the human population is the inequality of vaccine access due to the absence of proper health infrastructure or simply the high price of an individual dose [29]. In this regard, the selected liposomal formulation would be-low cost and relatively fast to prepare. The thin-film layer rehydration method is not ideal for industrial upscaling, as it is meant for small batch formulation. However, the formulation process can be translated to a more upscalable method such as microfluidics with little optimization [30].

The requirement for an ultra-low cold chain is a major drawback for vaccine availability and is extremely costly [31]. Compared to other SARS-CoV-2 vaccines already on the market, such as mRNA vaccines, a major advantage of DNA vaccines is their stability at 4 °C, which would improve the availability of the vaccine and ease its transport and related costs. When compared to licensed viral-vectored vaccines, there is a risk of developing neutralizing antibodies against the viral vector, thus reducing the efficacy of the second dose of vaccine [32]. Hence, this vaccine platform requires a longer interval (8 weeks or more) between each dose to achieve high efficacy [33]. This might not be suitable for an urgent vaccination campaign in a situation marked by a high incidence of infected cases. Moreover, plasmid DNA itself contains unmethylated CpG sequences that could activate innate immunity and enhance Th1 response. Killed vaccines are often formulated with aluminum salt, a Th2-bias adjuvant. This type of response has been associated with an enhanced lung pathogenesis following immunization with killed vaccine against SARS-CoV-1 [34], RSV [35], and measles virus [36]. Thus, by using a DNA vaccine, the risk of side effects caused by Th2-bias response is very unlikely. Moreover, viral vectors require highly regulated biosafety laboratories in order to handle live viruses. The necessity of these facilities could be a problem in some developing countries and could prevent them from developing their own manufacture chains, thus slowing down vaccine distribution.

Currently, Inovio is conducting a phase II clinical trial with its SARS-CoV-2 DNA vaccine candidate, INO-4800 [17]. INO-4800 is administered by intradermal electroporation through a CELLECTRA^®^ portable device. CELLECTRA^®^ is a great tool to facilitate electroporation administration. However, this technology still requires specific gear for vaccine delivery as well as specifically trained personnel, which add additional costs in a pandemic. To date, several DNA vaccines currently investigated in clinical trials have been shown to be immunogenic, safe, and well-tolerated [37]. Although concern over integration into the host genome has been raised, studies have found that the occurrence rate is lower than spontaneous mutations, and no evidence of such integration has been noted [38]. More importantly, an excellent safety record was also documented, even when different plasmid backbones and inserted genes were investigated [39].

In this study, as shown in the results, DOTAP4 used at N/P ratios of either 1:1 or 0.25:1 enhanced the immunogenicity of pCMVkan-S when compared to naked plasmid injection. More importantly, antibody responses both in lung and in serum induced by DOTAP4/pCMVkan-S formulations were comparable to those delivered by electroporation. In addition, the analysis of IgG isotypes showed balanced Th1/Th2 responses similar to electroporation. In agreement with the cytokine secretion profile, DOTAP4/pCMVkan-S formulations did not enhance IL-4 secretion (Appendix A). This might imply that DOTAP4/pCMVkan-S formulations could avoid the concern of Th2 bias response that led to pathogenesis [10].

In general, although DNA vaccines induce lower antibody responses when compared to other vaccine platforms such as mRNA or live-viral vectors, previous studies showed that macaques immunized with SARS-CoV-2 DNA vaccine developed a similar level of neutralizing antibody when compared to the titers found in human convalescent sera [15]. Unfortunately, in this study, IgA was undetectable both in lung and in serum at week 6, which was probably due to IgA levels usually being induced at lower magnitude and declining very rapidly compared to IgG [40].

However, as the T cell responses induced by DOTAP4/pCMVkan-S formulations were still lower than measured for electroporation, further modification and optimization of the formulations is required. In this study, we immunized mice with 100 µg of DNA, which is a relatively high dose for mice. We further evaluated the 100-fold dose reduction to 1 µg/dose. The similar trends of response that DOTAP4/pCMVkan-S at both N/P ratios showed were higher neutralizing antibody titers than observed with IM injection of naked plasmid. The N/P ratio of 0.25:1 tended to induce higher responses than the ratio of 1:1. However, mice immunized with IM-EP showed the highest responses (Appendix A).

The reason why 0.25:1 induced a stronger response than a 1:1 ratio, especially in terms of T cell activation, remains unclear, particularly considering that the amount of free DNA in the 0.25:1 formulation was more important compared to the other formulation (see Figure 5). Higher toxicity of 1:1 N/P ratio compared to 0.25:1 can be excluded, as mitochondrial activity assays were performed on RAW 264.7 cells at in vivo dosage, and after 24 h incubation with lipoplexes, no decrease in activity was observed compared to non-treated cells (see Appendix A). Release kinetics may play a role in this regard; in Figure 7, free pCMVkan-S was shown to start degrading after 15 min incubation in a protein-rich medium. However, this phenomenon was not observed when complexes were formed, where also DNA detached from complexes was still intact after incubation. This suggests that weak chemical interactions binding DNA and liposomes are still protecting the nucleic acid from degradation of proteins. At 0.25:1 and 1:1 ratios, liposomes do not present sufficient positive charges to form chemical bonds as strong as those observed in electrophoretic mobility studies for 25:1 to 100:1 ratios. Therefore, a faster release of DNA was observed when complexes were put under electrophoretic stress. This property of protecting pCMVkan-S from degradation and, at the same time, easily releasing DNA when subjected to a stressful environment, as the lysosome, could explain why lower N/P ratios resulted in better efficacy [41]. Moreover, a stronger interaction of pCMVkan-S with the liposomes could inhibit protein expression instead of enhancing it. Heidari et al. showed how anionic lipoplexes, with a DNA excess, could induce higher immunogenic responses in in vivo studies when compared to cationic lipoplexes [42].

There were some limitations to this study. The protective efficacy of immunized mice could unfortunately not be assessed in this study due to limited access to an animal biosafety level 3 facility. Further investigations could include challenge studies using more relevant models such as hACE2 transgenic mice, non-human primates, or ferrets [43] in order to assess elicited protective effect. However, this study provides initial information that at a 100 µg dose, the formulations could enhance the antibody responses at a level similar to electroporation, which is currently being evaluated in a clinical trial.

## 5. Conclusions

Stable cationic liposomes were formulated and complexed to negatively charged pCMVkan-S DNA vaccine expressing for SARS-CoV2 S protein. In vivo studies on mice showed an antibody response and neutralizing antibody titers similar to IM-EP. The T cell response in IM-EP was still higher than the DOTAP4/pCMVkan-S formulations; however, significantly higher titers for the lipoplex formulations were observed when compared to mice injected with pCMVkan-S alone. Further studies, such as challenge studies on relevant animal models, or on translation of the formulation method to a scale-independent manufacturing method, still need to be conducted. Nonetheless, the results of our study are promising, and this vaccine could present a valuable accessible alternative to current vaccines in low-income countries.

## Figures and Tables

**Figure 1 vaccines-09-00874-f001:**
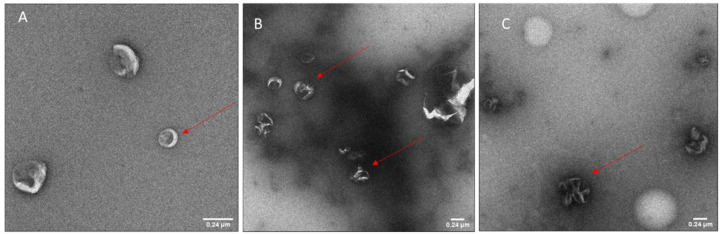
TEM images of blank DOTAP4 liposomes (**A**), DOTAP4-pCMVkan-S 0.25:1 N/P ratio (**B**) and DOTAP4-pCMVkan-S 1:1 N/P ratio (**C**). Bar = 0.24 µm. Red arrows indicate the presence of liposomes in figure (**A**), and of lipoplexes in figure (**B**,**C**).

**Figure 2 vaccines-09-00874-f002:**
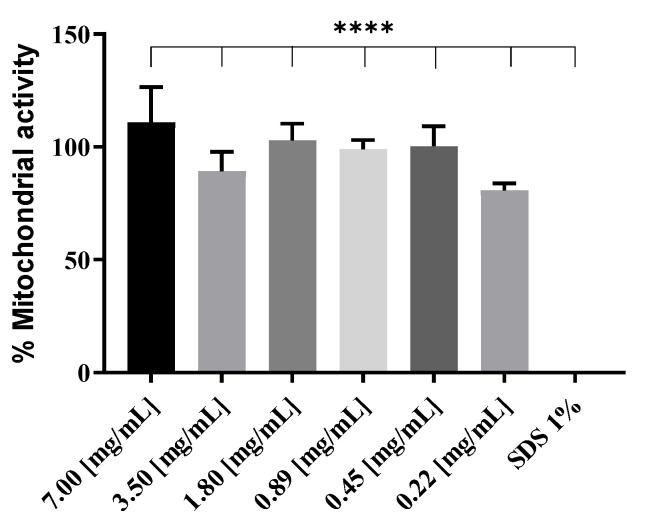
Effect on mitochondrial activity of different concentrations of blank DOTAP 4 liposomes on RAW 264.7 cells after 4 h incubation. Data were normalized to non-treated cell results; SDS served as positive control. Data are presented as mean ± SD (*n* = 3). **** indicate *p* < 0.0001 when comparing SDS 1% group with other groups.

**Figure 3 vaccines-09-00874-f003:**
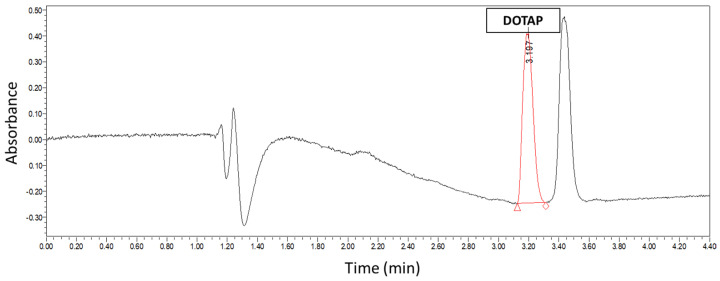
U-HPLC chromatograph of DOTAP4 sample. DOTAP peak in red and DOPE peak to the right. Liposome concentration was calculated by measuring the AUC of DOTAP peak.

**Figure 4 vaccines-09-00874-f004:**
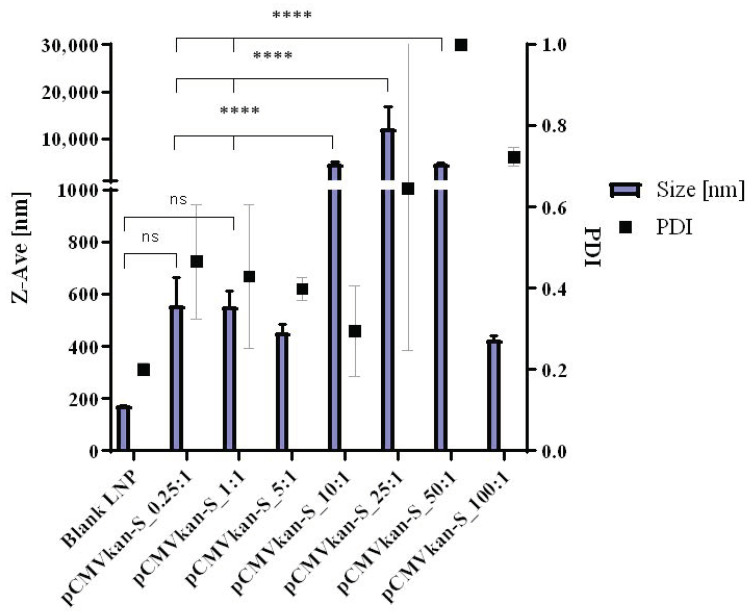
DLS results for DOTAP4/pCMVkan-S complexes at different N/P ratios, Z-Ave [nm], and PDI are plotted and jointly analyzed in this graph. Results are compared to blank liposome size profile. Lipoplex data were measured 30 min after liposome-DNA complexation at various N/P ratios. Data are presented as mean ± SD. **** indicates *p* < 0.0001, ns = *p* > 0.05, not statistically significant when compared to other groups.

**Figure 5 vaccines-09-00874-f005:**
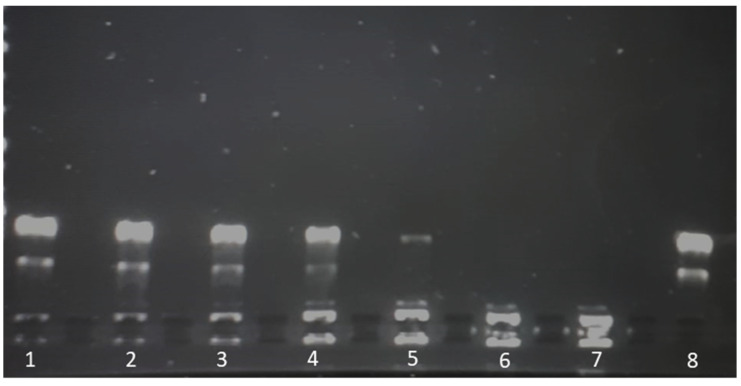
1% Agarose gel electrophoresis of pCMVkan-S in PBS: DOTAP4/pCMVkan-S N/P ratio 0.25:1 (1), 1:1 (2), 5:1 (3), 10:1 (4), 25:1 (5), 50:1 (6), 100:1 (7), and naked pCMVkan-S (8). Migration was performed in TBE buffer at 80 V for 30 min.

**Figure 6 vaccines-09-00874-f006:**
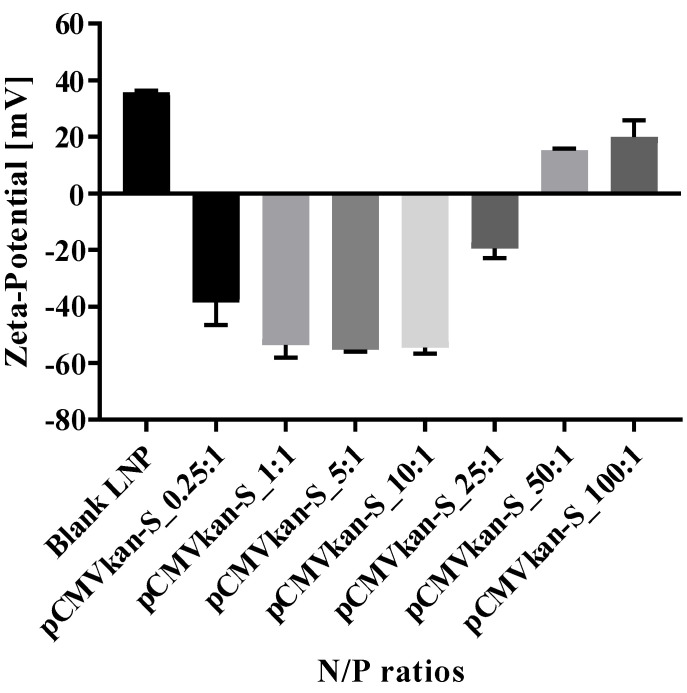
Zeta-potential results for DOTAP4/pCMVkan-S complexes at different N/P ratios (*n* = 6). Results are compared to blank liposome size profile. Lipoplex data were measured 30 min after liposome-DNA complexation at various N/P ratios.

**Figure 7 vaccines-09-00874-f007:**
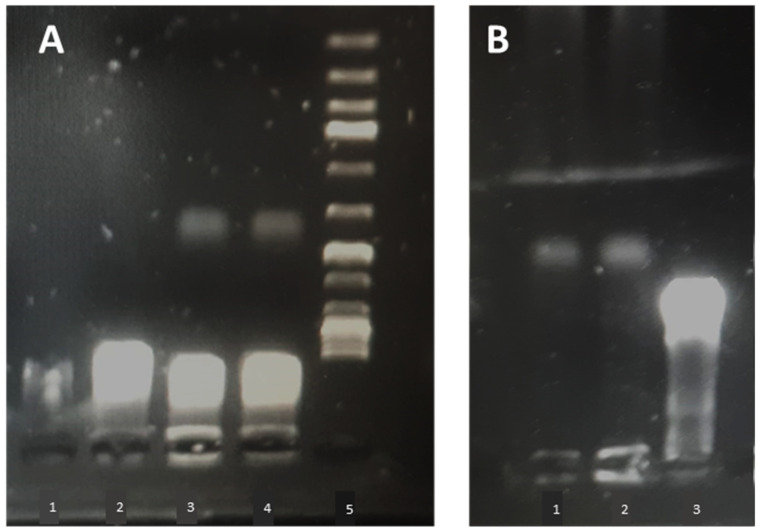
(**A**) Agarose gel electrophoresis of pCMVkan-S in C-MEM (1), pCMVkan-S in PBS (2), DOTAP4/pCMVkan-S N/P ratio 1:1 (3) and 0.25:1 (4), and DNA ladder of 1 kbp at RT after 15 min dilution in C-MEM (5). (**B**) Agarose gel electrophoresis of DOTAP4/pCMVkan-S N/P ratio 0.25:1 (1) and 1:1 (2) and pCMVkan-S diluted in PBS (3), after 37 °C, 24 h incubation.

**Figure 8 vaccines-09-00874-f008:**
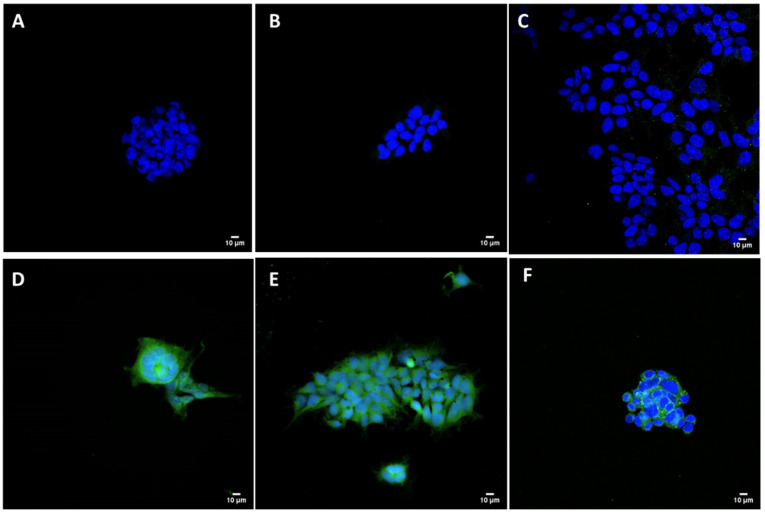
Intracellular SARS-CoV-2 S protein expression. HEK293 cells were transfected with pCMVkan-S alone or as nanoplexes. (**A**) = cells alone; (**B**) = only secondary Ab; (**C**) = pCMVkan-S alone; (**D**) = N/P 0.25:1 DOTAP4/pCMVkan-S; (**E**) = N/P 1:1 DOTAP4/pCMVkan-S; (**F**) = lipofectamine/pCMVkan-S lipoplexes. Green = S protein, Blue = nucleus. Images were obtained by confocal laser scanning microscopy, ×40 magnification, bar = 10 µm.

**Figure 9 vaccines-09-00874-f009:**
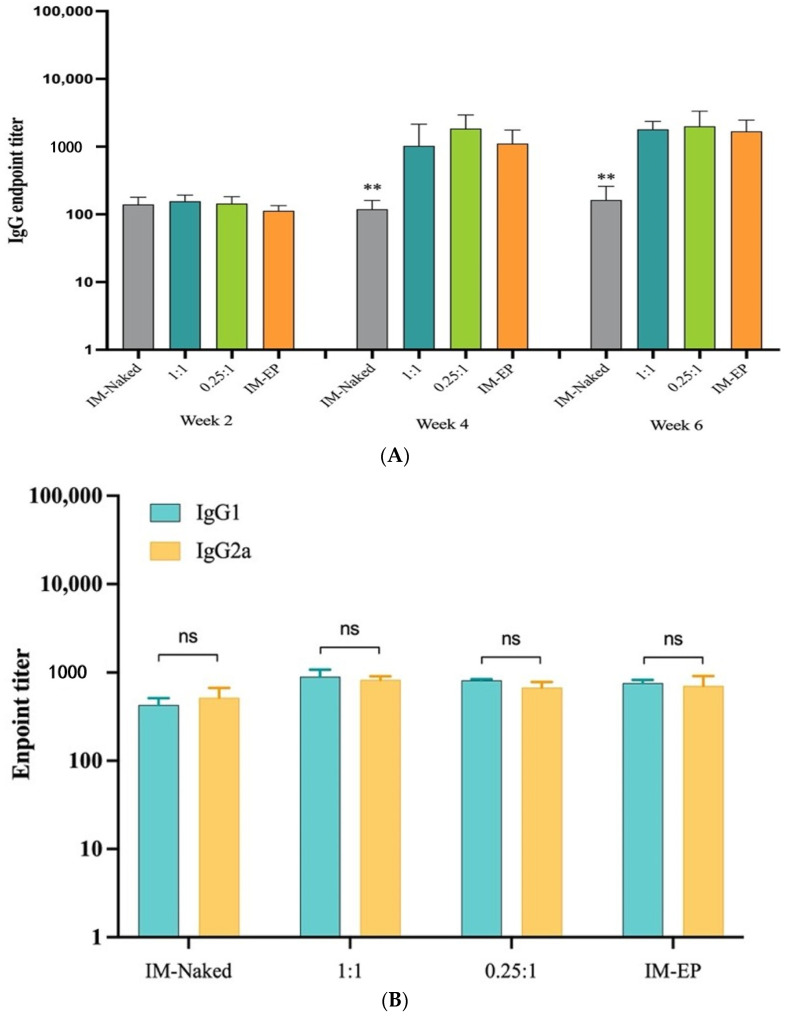
(**A**) Endpoint titers of SARS-CoV-2 spike-specific total IgG analyzed at weeks 2, 4, and 6 in mice immunized intramuscularly with naked pCMVkan-S (IM-naked), pCMVkan-S formulated with DOTAP4 at N/P ratios of 1:1 and 0.25:1, and by using electroporation device (IM-EP), (**B**) IgG subclass analysis; IgG1 and IgG2a in immunized mice sera collected on week 6. Data presented as mean ± SD of the endpoint titers in each vaccination group (*n* = 5). ** indicate *p* < 0.01 when compared to other groups. ns: not statistically significant.

**Figure 10 vaccines-09-00874-f010:**
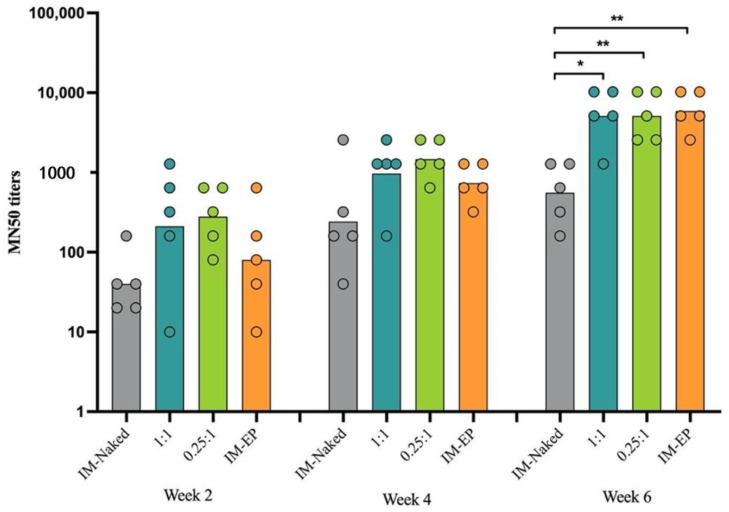
Neutralizing antibody analysis by microneutralization (MN) assay. Bars represent the geometric mean of the MN50 titers (GMT) from mice immunized intramuscularly with naked pCMVkan-S (IM-naked), pCMVkan-S formulated with DOTAP4 at N/P ratios of 1:1 and 0.25:1, and by using electroporator device (IM-EP), *n* = 5. * and ** indicate *p* < 0.05 and *p* < 0.01, respectively.

**Figure 11 vaccines-09-00874-f011:**
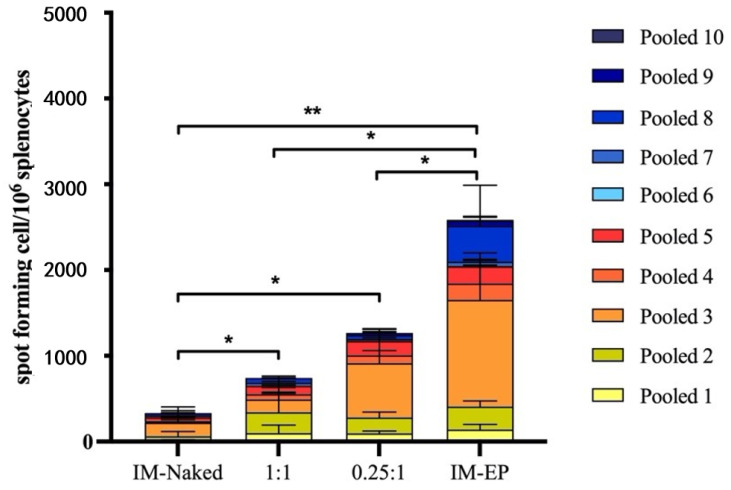
SARS-CoV-2 spike-specific T cell responses analyzed by ELISpot. Mouse splenocytes were stimulated with pooled peptides from S1 (pool 1–5) and S2 (pool 6–10) regions. Each bar represents sum of IFN-γ responses in mice (*n* = 5) immunized with different immunization strategies including naked pCMVkan-S (IM-naked), pCMVkan-S formulated with DOTAP4 at N/P ratios of 1:1 and 0.25:1, and by using electroporator device (IM-EP). * and ** indicate *p* < 0.05 and *p* < 0.01, respectively.

## Data Availability

Data are also available in the supplementary materials and under reasonable request to the corresponding authors.

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
