# Peer review of "DNA Vaccine Administered by Cationic Lipoplexes or by In Vivo Electroporation Induces Comparable Antibody Responses against SARS-CoV-2 in Mice"

_vaccines, 2021, doi:10.3390/vaccines9080874_

Round 1

Reviewer 1 Report

In this manuscript, the authors describe the characterization of a lipid formulation for their recently developed DNA vaccine. Overall, they show that the liposome formulation does not underperform compared to the traditional electroporation method in mice (excepting for IFNg responses). It is somewhat underwhelming in the absence of a viral challenge protection experiment and arguable whether they should state that it "enhances immunogenicity" when it performs about equally. 

Specific comments:

  • Reconsider the title with regards to enhancing the immunogenicity of the vaccine
  • In the Results section, consider adding slightly more narrative prior to jumping into the results to improve readability. For example, "We characterized the liposomes by X/Y/Z..." Similarly, a brief into to the WST toxicity assay could help and might put the results in context (We expect to see 70-100% activity in the absence of toxicity...)
  • Why are you presenting Fig 6 as #6 if you're mentioning first? Consider either re-ordering the figures or moving the statement regarding TEM to later.
  • Figure 1, please use "." instead of "," in the concentrations.
  • Consider using color in Fig 9-10 and especially in Fig 11 (similar to the PLOS manuscript). Particularly for Fig 11, color and/or patterns might help improve legibility. The pools are difficult to see in grayscale with the error bars.
  • Line 504, please clarify the statement that "current SARS-CoV-2 S gene delivery in licensed vaccines require viral vectors" -- the mRNA vaccines do not.

Author Response

Q1: In this manuscript, the authors describe the characterization of a lipid formulation for their recently developed DNA vaccine. Overall, they show that the liposome formulation does not underperform compared to the traditional electroporation method in mice (excepting for IFNg responses). It is somewhat underwhelming in the absence of a viral challenge protection experiment and arguable whether they should state that it "enhances immunogenicity" when it performs about equally. 

A1: Thanking the reviewer for their comments, we would like to emphasize that the liposomal formulation enhanced the immunogenicity in comparison to the "naked" plasmid. This is promising, as the use of electroporation for the application of a vaccine requires specialized equipment, which may not be available at sufficient quantities, as well as additional training of personnel. Undoubtedly, electroporation would be more labor- and cost-intensive, two factors of high concern in a low/medium income country. Unfortunately, due to the current situation in Thailand, a challenge study could not be performed at this point but is on the agenda going forward.

Specific comments:

Q2: Reconsider the title with regards to enhancing the immunogenicity of the vaccine

A2: To better reflect its contents, the title of the manuscript has been changed to:

DNA vaccine administered by cationic lipoplexes or by in vivo electroporation induce comparable antibody responses against SARS-CoV-2 in mice”

Q3: In the Results section, consider adding slightly more narrative prior to jumping into the results to improve readability. For example, "We characterized the liposomes by X/Y/Z..."

A3: We added an introductory paragraph and more specific explanations of the results, especially regarding the characterization part. We are sure that this will help improve readability. (Lines 293-298)

Q4: Similarly, a brief into to the WST toxicity assay could help and might put the results in context (We expect to see 70-100% activity in the absence of toxicity...)

A4: The following has been added to the manuscript: “WST-1 toxicity results showed absence of toxicity on RAW 264.7 cells after 4 h incubation and up to a concentration of 7 mg/mL of blank liposomes (Fig. 2). An incubation time of four hours was chosen as liposomes are estimated to be up taken by the cells during this time period. Data were normalized to results obtained with non-treated cells representing 100% of mitochondrial activity. For all sample conditions mitochondrial activity was >79%. This difference is not significant when compared to the non-treated sample, showing the absence of cytotoxicity of the samples at high concentrations. As observed in Fig. 2, incubation with SDS 1% (positive control) resulted in complete loss of mitochondrial activity.‘’ cf. line 312-320

Q5: Why are you presenting Fig 6 as #6 if you're mentioning first? Consider either re-ordering the figures or moving the statement regarding TEM to later.

A5: The order of the figures was corrected.

Q6: Figure 1, please use "." instead of "," in the concentrations.

A6: This was corrected.

Q7: Consider using color in Fig 9-10 and especially in Fig 11 (similar to the PLOS manuscript). Particularly for Fig 11, color and/or patterns might help improve legibility. The pools are difficult to see in grayscale with the error bars.

A7: Colors are now used in Fig. 9-11.

Q8: Line 504, please clarify the statement that "current SARS-CoV-2 S gene delivery in licensed vaccines require viral vectors" -- the mRNA vaccines do not.

A8: This part of the discussion was revised, please refer to lines 544-551.

Reviewer 2 Report

In the manuscript by Peletta et al., the authors describe a cationic liposome formulation for a SARS-CoV-2 vaccine regimen.  The manuscript builds on the authors’ previous work regarding a DNA vaccine candidate.  The results are interesting because they show how already studied vaccines can be improved.  However, the presentation of the results is not clear and makes review of the manuscript difficult.  Also, the authors fail to make the proper comparisons to already in-use vaccine regimens that utilize similar liposome technology.  Below I outline areas where the manuscript must be improved.

Title:  the results and conclusions presented in the manuscript have nothing to do with a more equitable vaccine distribution.  Thus the title is misleading.

The very first figure call-out in the results is “Fig. 6” in line 292.  Why call out data that is not presented until lines 338-339?  Furthermore, what exactly is being shown in Figure 6?  Please annotate the figure panel with arrows showing the reader what the authors describe in the text and figure legend.

Figures 1, 3, 5, and 9B do not include any statistical tests.

The captions of Figures 1-6 include only the caption title and provide no detailed description of the results shown and how to properly interpret the data.  Moreover, the text also fails to properly describe the results, leaving it up to the reader to discern what the results are intending to show.  For example, in line 317, “showed complex stability in electrophoresis studies.”  Many readers may not know how to interpret these electrophoresis studies.

Figure 2 – it is not clear how the plot shown relates to what is written in lines 300-307.

The authors must do a better job of relating their vaccine design to regimens that are already in use and utilize liposomes for delivery.  These vaccines work very well, so it is not clear how the authors’ vaccine design improves what already works great.

Author Response

Comments and Suggestions for Authors

In the manuscript by Peletta et al., the authors describe a cationic liposome formulation for a SARS-CoV-2 vaccine regimen.  The manuscript builds on the authors’ previous work regarding a DNA vaccine candidate.  The results are interesting because they show how already studied vaccines can be improved.  However, the presentation of the results is not clear and makes review of the manuscript difficult.  Also, the authors fail to make the proper comparisons to already in-use vaccine regimens that utilize similar liposome technology.  Below I outline areas where the manuscript must be improved.

Q9: Title:  the results and conclusions presented in the manuscript have nothing to do with a more equitable vaccine distribution.  Thus the title is misleading.

A9: To better reflect its contents, the title of the manuscript has been changed to:

“DNA vaccine administered by cationic lipoplexes or by in vivo electroporation induce comparable antibody responses against SARS-CoV-2 in mice”

Q10: The very first figure call-out in the results is “Fig. 6” in line 292.  Why call out data that is not presented until lines 338-339?  Furthermore, what exactly is being shown in Figure 6?  Please annotate the figure panel with arrows showing the reader what the authors describe in the text and figure legend.

A5: The order of the figures was modified for better readability. More text on this figure was added (lines 309-311) and arrows were added to improve understanding.

Q11: Figures 1, 3, 5, and 9B do not include any statistical tests.

A11: In former Fig. 1 and 3 (now Fig. 2 and 4) statistical analysis was added. In figure 2 the important statistics is the significant difference with the ‘’non-treated cell’’ sample onto which all the other results were normalized. We explain in the text that there is no significant difference between the samples and the ‘’non-treated cells’’ control, which is the information we need to state that the sample is non- toxic. Statistical analysis compared with positive control was added.

The aim of Fig. 6 (previously 5) is to show the Zeta-potential switch from negative to positive and compare this to the size and electrophoretic mobility graph (Fig. 4, 5), therefore we think statistical analysis is not needed. Statistical test for Fig 9B was also included.

We also added information on statistical analysis method in lines 288-291.

Q12: The captions of Figures 1-6 include only the caption title and provide no detailed description of the results shown and how to properly interpret the data.  Moreover, the text also fails to properly describe the results, leaving it up to the reader to discern what the results are intending to show.  For example, in line 317, “showed complex stability in electrophoresis studies.”  Many readers may not know how to interpret these electrophoresis studies.

A12: Additional information was added to the captions. The text describing Fig. 1-6 was completely revised.

Q13: Figure 2 – it is not clear how the plot shown relates to what is written in lines 300-307.

 A 13: Details were added to former Fig. 2 (now Fig. 3) and some text revision was made. Cf. line 330-338.

Q14: The authors must do a better job of relating their vaccine design to regimens that are already in use and utilize liposomes for delivery.  These vaccines work very well, so it is not clear how the authors’ vaccine design improves what already works great.

A14: In accordance with the reviewer's suggestion, we added more information to the Discussion. Overall, the aim of the work was to propose an easily manufacturable and low cost alternative to existing vaccines that proved to be very effective but also fairly expensive (either in manufacture or transport/storage). The objective would be to perform a tech transfer to Thailand to allow our collaborators to produce the vaccine and delivery system locally.

Round 2

Reviewer 2 Report

The author's have adequately addressed my previous concerns.